# The Effect of Parenting Peer Education Interventions for Young Mothers on the Growth and Development of Children under Five

**DOI:** 10.3390/children10020338

**Published:** 2023-02-09

**Authors:** Dewi Rokhanawati, Harsono Salimo, Tri Rejeki Andayani, Mohammad Hakimi

**Affiliations:** 1Public Health Department, Faculty of Medicine, Universitas Sebelas Maret, Surakarta 57126, Indonesia; 2Department of Midwifery, Faculty of Health Sciences, Universitas ‘Aisyiyah Yogyakarta, Yogyakarta 55292, Indonesia; 3Department of Psychology, Faculty of Psychology, Universitas Sebelas Maret, Surakarta 57126, Indonesia; 4Department of Public Health, Faculty of Medicine, Universitas Gadjah Mada, Yogyakarta 55281, Indonesia

**Keywords:** parenting peer education, parenting self-efficacy, parenting behaviour, young mothers, growth and development of children under five

## Abstract

One of the contributing factors in the growth and development of children under five is the mother’s ability to provide childcare, but young mothers do not have enough parenting skills. The goal of the current study was to examine the effect of the parenting peer education (PPE) programme on young mothers’ parenting self-efficacy and behaviour, and the growth and development of children under five. There were two groups, which were a control group (without intervention) and an intervention group, in which there were 15 participants in each group. Analysis covariance with the pre-test scores as covariates was used in this study. The results showed that, compared with the control group, the intervention group reported significantly better parenting self-efficacy, parenting behaviour, children’s growth, and children’s development, including cognitive, language, and motoric aspects. The PPE programme can exchange the young mothers’ experiences on how their children grow and develop, and the mothers will also receive psychological support. In conclusion, the PPE programme affected the young mothers’ parenting self-efficacy and parenting behaviour and the children’s growth and development.

## 1. Introduction

The under-five period is a very important period for children because the fastest development and growth rates occur in this period. According to the World Health Organization (WHO), 7.3% of children under five are malnourished, 5.9% of children under five are overweight, and 21.9% of children under five have stunted growth [1]. The children’s development and growth rates in developing countries, including Indonesia, are slower than in developed countries due to the lack of consumption of food or nutrition in developing countries. If the problem can be addressed early on, Indonesian children can grow and develop properly [2].

The process of children’s development is greatly influenced by the role of parents [3]. The parenting quality influences the growth and development of children [4]. Good parenting supported by good communication can improve the children’s development, while poor parenting can result in a negative impact on the children’s development [5,6]. Interventions in parenting can improve the quality of parenting behaviour towards children, the quality of children’s health, and the healthy bond between children and their parents [7,8,9]. One of the challenges that parents face in parenting is the lack of parenting knowledge and no special preparation for becoming parents, because the parents do not have enough parenting knowledge and skills because they are too young when they get married [10].

The high cases of teenage marriage, especially in developing countries such as India (26%), Myanmar (16%), the Philippines (15%), Sri Lanka (12%), and Indonesia (11%), contribute significantly to the number of cases of marriage at a young age, <18 years, in the world [11]. Teenage marriage is found in geographical pockets throughout Indonesia, with rates varying widely across the country and by the level of government (province, district and sub-district). In 2018, approximately 11% or one in nine women aged 20–24 were married before the age of 18, and approximately 1% or one in one hundred men aged 20–24 were married before the age of 18. It is estimated that there are 1,220,900 girls married under 18 years old and a 0.56% prevalence of women aged 20–24 who were married before the age of 15 [12]. Early marriages are still common in Bangladesh (69%), Nepal (52%), India (41%), and Pakistan (37%) [13].

Teenage marriage is mostly experienced by young women who live in rural areas with low levels of parental education and who do not have access to media information [13,14]. Teenage marriage can cause babies born to die within the first 28 days of birth, low birth weight, and complications during pregnancy and childbirth [12]. The case studies from four countries—Bangladesh, Guatemala, Ethiopia, and Kenya—shed light on the shared underlying factors that drive adolescent girls’ risk of teenage marriage, pregnancy, and premature school leaving [15].

DeVito [16] studied a descriptive-correlation of adolescent mothers 4–6 weeks postpartum at four postpartum clinical sites in northern and central New Jersey. The results of the study showed that most teenage mothers needed guidance and support from health workers in becoming parents. Based on educators’ understanding of adolescent mothers’ experiences of parenting, childbirth and caring for infants, educators can develop additional programmes, activities, and educational opportunities that can assist adolescent mothers with their goals and aspirations in the future [16]. A study meta-analysis by Gibbs et al. reported that experiencing childbirth at chronological age ≤16 years had an influence on women’s health and their baby’s health. The results of the study showed that young maternal age increased the risk of maternal anaemia quite strongly, although information on other nutritional outcomes and maternal morbidity/death was less clear. Many of the differences observed among older adolescents with respect to infant outcomes were likely due to socio-economic or behavioural differences from country to country or setting [17]. A study by Yu et al. in Africa, Asia, and Latin America used design cross-sectional data from results of Demographic Health Surveys from 18 countries to select the first-born child of mothers aged 15–24 years and a range of potential confounding factors, including maternal height. The study reported that there were about twice as many cases of stunting in Africa, where there was an increase of 10 ppts in stunting for children of young mothers [18].

According to Miller [19], parenting is a shared responsibility between father and mother. However, not all fathers have enough time to help with childcare, so that most of the childcare is borne by mothers. Unfortunately, some mothers do not have the ability to raise children due to their young age. Women with low education usually marry at a young age, so they are not psychologically ready to contribute to the growth and development of their children [20]. In addition, the limited skills in the parenting of young parents and the double burden that must be borne by the young mother result in the poor growth and development of their children [21]. The mother’s low education level is related to low socioeconomic level [22]. According to Jacquet et al. [23], successful parenting of children under three years in families with low socioeconomic levels requires special attention and support in the form of providing affordable and easily accessible care services.

The successes of parents in parenting are also supported by optimal self-efficacy. Parenting self-efficacy describes a parent’s belief in their ability to perform the parenting role successfully. Higher levels of parenting self-efficacy have consistently been shown to be correlated with a wide range of parenting and child outcomes [24]. Elliot et al. [25] stated that mothers who have good parenting self-efficacy will prioritize the needs and welfare of their children and are willing to sacrifice for their children. According to Vance et al., parenting self-efficacy is known to contribute to infant development and health status [26]. Mouton et al. stated that a parenting programme that focuses on improving self-efficacy is effective in reducing child externalizing behaviour [27]. The study by Ruiz-Zaldibar et al. showed that parenting programmes that promote parental competency in healthy lifestyles provide promising results for increasing self-efficacy and parenting styles [28]. The levels of self-efficacy and parenting behaviour are influenced by several factors, including age, education, previous experience, positive family environment, information support, and culture [29,30,31,32,33].

The parenting programme for young mothers is one of the methods that can be used to improve their quality as parents in their family, through the instilling of self-efficacy, friendly parenting behaviour in education, nutrition, nurturing, and protection needed by their children, which will optimize children’s growth and development rates according to the children’s age. Based on our literature review, we found that some previous studies have studied the application of parenting programmes for young mothers, which are the home visit programme [34], the stimulation programme [35], and the Mindful with Your Baby programme [36]. In detail, we summarized the previous studies and present them in Table 1. Those previous studies did not simultaneously study the effect of the parenting programmes on the young mothers’ parenting self-efficacy, the young mothers’ parenting behaviour, and the growth and development of children under five. One of the potential parenting programmes for young mothers is the peer parenting education (PPE) programme. The advantage of this programme is the provision of parenting education by peers. Adolescence, as a period of transition from childhood to adulthood, has special characteristics such as searching for identity, rapid growth and development, and mental and emotional maturity that is still unstable [37,38]. These special characteristics are likely to be understood by other youth of the same age. Based on Table 1, the PPE programme has not been applied to young mothers yet.

The PPE programme uses the general basic principles of a peer education programme. Peer education is a popular concept that implies an approach, a communication channel, a methodology, a philosophy, and a strategy [39]. Peer education is now viewed as an effective behavioural change strategy, and it draws on several well-known behavioural theories—Social Learning Theory, the Theory of Reasoned Action and the Diffusion of Innovation Theory. Peer education has been used in many areas of public health, including nutrition education, family planning, substance use, teenage pregnancy prevention, adolescent HIV-AIDS knowledge and self-efficacy, and violence prevention [39,40,41]. Furthermore, in this study, peer education in parenting (called parenting peer education (PPE)) is applied for young mothers. In most societies, it is often difficult for young mothers to obtain clear and correct information on issues such as reproductive health and child growth and development. In the PPE programme, community members are trained as peer-to-peer educators to provide interactive education/training on how to raise children, monitoring the growth and development of toddlers using guidebooks. In particular, the PPE programme has the goal of providing knowledge and skills to young mothers about childcare, in order to achieve optimal child growth and development [39].

The PPE programme is a process of peer education for young mothers raising and/or caring for children under five, which takes place over 3 months by measuring the young mothers’ parenting self-efficacy (PSE), young mothers’ parenting behaviour, children’s growth, and children’s development. The measurement of parenting self-efficacy (PSE) can be conducted by using a questionnaire which is translated and adapted from Van Rijen et al. [42]. Meanwhile, the measurement of parenting behaviour can be conducted by using a questionnaire which is translated and adapted from Van Leuwen and Vermut [43]. The growth of children can be measured by comparing the weight-for-age value with the Z-Score. The development of children (including cognitive, language, and motoric aspects) can be measured using the Bayle scale test [44].

The goal of this study was to examine the effect of the PPE programme on young mothers’ parenting self-efficacy (PSE), young mothers’ parenting behaviour, and the growth and development of children under five. Thus, this study was new and has not been conducted by the other authors yet. Furthermore, the hypotheses of this study were (1) participants in the intervention group will report significantly higher PSE than participants in the control group (without intervention) after the PPE programme; (2) participants in the intervention group will report significantly higher parenting behaviour than participants in the control group (without intervention) after the PPE programme; (3) participants in the intervention group will report significantly better children’s growth than participants in the control group (without intervention) after the PPE programme; (4) participants in the intervention group will report significantly better children’s development (cognitive, language and motoric aspects) than participants in the control group (without intervention) after the PPE programme. In this study, the measurement of parenting behaviour refers to the parenting behaviour scale instrument consisting of nine scales: positive parenting, supervision, rules, discipline, inconsistent discipline, harsh punishment, neglect, material rewards, and autonomy [43].

## 2. Materials and Methods

### 2.1. Participants

The participants were young mothers in the public health centres in Gunung Kidul District, Yogyakarta Province, Indonesia, and were selected through a screening process. A total of 30 eligible participants were recruited to be respondents in this study. Subjects in this study were divided into two groups, namely the intervention and control groups. The intervention group was given the intervention of the PPE programme using the PPE module and the help of a website-based PPE application so the participants carried out parenting behaviour. Meanwhile, the control group did not receive the intervention and only received standard services in the Health Centres. The inclusion criteria were: (1) young mothers aged <20 years; (2) young mothers whose first child is aged 12–42 months; (3) young mothers who have a minimum education level of Elementary School. The reason for selecting respondents was because the age below 20 years is a transitional age where young mothers have to play a double role, namely becoming parents as well as completing the unfinished developmental process towards adulthood.

The selection of the PPE programmes is based on the peer concept, where young mothers and peer educators are the same age, which allows them to be better able to communicate and have empathy for one another. The intervention also uses a web-based PPE application that makes it easier for peer educators and respondents in the intervention group to access information. The web-based application is easier to develop, easy to access using a variety of devices, easy to distribute information, and flexible. The PPE programme was conducted by peer educators (PEs) (as many as 2 young mothers) who were recruited based on the results of the screening. The peer educators (PEs) criteria were:young mothers aged <20 yearsthey have children aged 12 to 42 monthsthey have a minimum education of junior high school (graduated in basic education)they are interested and willing to take the time to become peer educatorsthey obtain their husband’s approvalthey fill out a screening questionnaire to become PEs.

After being selected as PEs, they were trained at the public health centres to increase the PEs’ knowledge about the PPE module and the use of the PPE applications.

### 2.2. Design and Procedure

This study was carried out using a quasi-experimental design. To prevent contamination of the treatment in the control group if using an experimental research design, we utilized a non-equivalent control group and a pre-test–post-test quasi-experimental design. The use of a quasi-experiment design considers that there is no randomization in the intervention and control groups, control of the variables that have an effect is not carried out because the research is carried out in the community, and the sample is limited/small [45,46]. In this study, the way to use the quasi-experimental design was to divide the sample into two groups, namely the intervention group and the control group. For the two groups, a pre-test was carried out. Furthermore, the intervention group was given the Conventional Health Services with the intervention of the PPE programme, and the control group received the Conventional Health Services. After the implementation of the intervention, a post-test was carried out in the two groups.

The population was 30 young mothers having children under 5 in the public health centres in Gunung Kidul District (Yogyakarta Province, Indonesia). The sample size was calculated using the test of differences between means from Lameshow et al. [47] so that samples in the control and intervention groups were 15 people in each group [47]. The intervention group was the group of young mothers with children under 5 with the PPE programme, and the control group was the group of young mothers with children under 5 without the PPE programme. A non-probability sampling technique with a purposive sampling approach in both groups, i.e., the intervention and control groups, was used in this study. Detailed steps in this study are shown in Figure 1. This research has passed the ethical feasibility test from the Health Research Ethics Committee of Sebelas Maret University (UNS) with No. 032/UN27.06.6.1/KEPK/EC/2020 (date: 7 February 2020).

### 2.3. Intervention

The PPE intervention was carried out for 3 months from September to December 2021. The PPE was carried out by two PEs recruited by researchers through a selection using the requirements shown in Section 2.1. At first, the PEs were trained using the PPE module. The training was carried out 2 times, with the implementation of research carried out during the COVID-19 pandemic. Thus, the intervention activities were carried out in a hybrid manner, namely a combination of online and offline. Face-to-face activities were carried out every 2 weeks by complying with the health protocol carried out in the halls of the Semanu I and Ponjong I Health Centres. The activities of all respondents and PEs were recorded in the PPE applications.

### 2.4. Measurement

The data in this study were collected using a questionnaire in the g-form and a direct examination of children under 5 to measure their growth and development. The details are described as follows:

#### 2.4.1. Parenting Self Efficacy

Parenting Self Efficacy (PSE) was measured using a questionnaire translated and adapted from Van Rijen et al. [42]. The translation and adaptation processes referring to the WHO standard included four steps, namely: (1) forward translation; (2) expert panel back-translation; (3) pre-testing and cognitive interviewing; (4) final version. PSE competency items assess the domains of nurturance, discipline, play, and routine. The process of translation and adaptation of the instrument was carried out by a person who had mastered two languages, namely English and Indonesian (an English lecturer with a master’s background, who was a member of the Language Institute of UNISA Yogyakarta). After that, each item in the instrument was examined by 3 experts, namely paediatricians, obstetricians and psychologists. These translation and adaptation processes of 26 items resulted in 17 items. Each item was rated with a Likert scale from 1 to 6, where 1 = strongly disagree, 2 = disagree, 3 = less agree, 4 = occasionally agree, 5 = agree, and 6 = strongly agree. The data obtained from the 30 young mothers in the Gunung Kidul district were tested for validity (Pearson’s product moment) and reliability (Cronbach alpha). The reliability test resulted in the Cronbach’s Alpha of 0.822.

#### 2.4.2. Parenting Behaviour

A parenting behaviour scale (PBS) was used to assess the parenting behaviour in this study. The PBS was translated and adapted from Van Leuwen and Vermust, which included positive parenting, supervision, rules, discipline, punishment, neglect, material rewards, and autonomy [43]. The process of translation and adaptation of the instrument was carried out by a person who had mastered two languages, namely English and Indonesian (an English lecturer with a master’s background, who was a member of the Language Institute of UNISA Yogyakarta). After that, each item in the instrument was examined by 3 experts, namely paediatricians, obstetricians and psychologists. The translation and adaptation process of 45 items resulted in 12 items. Each item was rated with a Likert scale from 1 to 5, where 1 = never, 2 = rarely, 3 = occasionally, 4 = often, and 5 = always. The data obtained from the 30 young mothers in the Gunung Kidul district were tested for validity (Person product moment) and reliability (Cronbach alpha). The reliability test resulted in the Cronbach’s Alpha of 0.805.

#### 2.4.3. Children under 5’s Growth

The measurement of the children under 5’s growth was based on the weight-for-age values which were compared with the Z-Score [48]. The Z-Score was calculated based on the weight-for-age value using the WHO Anthro application.

#### 2.4.4. Children under 5’s Development

The measurement of the children under 5’s development, including cognitive, language, and motoric aspects, can be conducted using the Bayley scale test [44]. The children’s development was measured using the Bayley Scales of Infant and Toddler Development Third Edition, or BSID-III test. The Bayley III test can be used for children aged 1–42 months. The Bayley III test aims to identify children’s development and intervention planning. The Bayley III test covered aspects of cognitive scale, language scale and motoric scale (fine and gross motoric skills). A child clinical psychologist who was from a team from Dr. Sardjito Hospital Yogyakarta carried out the Bayley III test. Bayley III tests were performed for 45–60 min for each child.

### 2.5. Data Analysis

All analyses were carried out using Stata software version 14 with Universitas Gadjah Mada (UGM) license. Before analysing the research data, normality and homogeneity tests were carried out for all research variables both in the intervention and control groups. The normality data test was carried out using the Shapiro–Wilk (SW) test (*n* < 50). The normality data test for the PSE resulted in SW = 0.89937, df = 1.322, *p* = 0.90738; for the PBS resulted in SW = 0.97360, df = −1.325, *p* = 0.90738; for children under 5’s growth resulted in SW = 0.93632, df = 0.417, *p* = 0.33829; for children under 5’s cognitive development resulted in SW = 0.98798, df = −2.881, *p* = 0.99802; for under 5 children’s language development resulted in SW = 0.94924, df = −0.032, *p* = 0.51261; for children under 5’s motoric development resulted in SW = 0.97436, df = −1.382, *p* = 0.91658. Furthermore, the homogeneity data test for young mothers’ parenting self-efficacy, young mothers’ parenting behaviour, children under 5’s growth, and children under 5’s development (cognitive, language, and motoric aspects) in the intervention and control groups showed a *p*-value > 0.05. Furthermore, analysis of covariance (ANCOVA) with pre-test scores as covariates was used for unbiased intervention estimates by assessing between-group differences. The *p*-value < 0.05 was considered a significant difference, and vice versa. Partial eta-squared (η^2^
*p*) was calculated to assess effect sizes where a value η^2^
*p* < 0.01 was considered a very small effect, 0.01 ≤ η^2^
*p* < 0.06 was considered a small effect, 0.06 ≤ η^2^
*p* < 0.14 was considered a medium effect, η^2^
*p* ≥ 0.14 was considered a large effect [49].

## 3. Results

### 3.1. The Distribution of Respondents’ Characteristics

The distribution of respondents’ characteristics is the basic characteristics of respondents obtained through the pre-test. The respondents’ characteristics in this study were young mothers’ parenting self-efficacy, young mothers’ parenting behaviour, children under five’s growth, and children under five’s development (cognitive, language, and motoric aspects) (Table 2).

Table 2 shows that the young mothers’ parenting self-efficacy score in the control group was higher than that in the intervention group. Furthermore, the young mothers’ parenting behaviour score was not significantly different in both groups. The children under five’s growth score was higher in the intervention group than that in the control group. In addition, the cognitive and language development scores of children under five were higher in the intervention group, while the motoric development score of children under five was higher in the control group.

### 3.2. The Normality Test for the Variable Data of Young Mothers’ Parenting Self-Efficacy and Behaviour, and the Growth and Development of Children under Five

In this study, the data normality was analysed through the Shapiro–Wilk test (*n* < 50). The results of the normality test for the variable data of young mothers’ parenting self-efficacy and behaviour, and the growth and development of children under five (cognitive, language, motoric aspects), in pre-test and post-test, in the control and intervention groups, showed the *p*-values above 0.05 (Table 3). Therefore, it means that the data of the variables were distributed normally.

### 3.3. The Homogeneity Test for the Variable Data of Young Mothers’ Parenting Self-Efficacy and Behaviour, and the Growth and Development of Children under Five

Before selecting the statistical test to be used in this study, a homogeneity test was conducted to test whether or not the relationship between the variances of two or more data distributions was homogeneous. The results of the homogeneity test for the variable data of young mothers’ parenting self-efficacy and behaviour, and the growth and development of children under five (cognitive, language, motoric aspects) in the control and intervention groups showed the *p*-values above 0.05 (Table 4). Therefore, it can be concluded that the data were normally distributed and homogeneous, so the statistical test used was the ANCOVA with the pre-test scores as covariates.

### 3.4. Analysis of Covariance (ANCOVA) with the Pre-Test Scores as Covariates

The *p*-values and partial eta-squared values obtained from the ANCOVA with the pre-test scores as covariates are shown in Table 5.

Table 5 showed that, compared with the control group, the young mothers in the intervention group reported significantly better parenting self-efficacy (*p* = 0.000, η^2^ *p* = 0.308), parenting behaviour (*p* = 0.015, η^2^ *p* = 0.256), children’s growth (*p* = 0.000, η^2^ *p* = 0.307), children’s cognitive development (*p* = 0.001, *η2 p* = 0.521), and children’s language development (*p* = 0.032, η^2^ *p* = 0.081). Meanwhile, the PPE programme resulted in a statistically significant difference in children’s motoric development (*p* = 0.046) between the control group and intervention group, but the effect size was very small (η^2^ *p* = 0.008). It means that children under five’s motoric development score between the control group and the intervention group was not much different after the PPE programme was given to the young mothers.

## 4. Discussion

This study examined the effect of the parenting peer education (PPE) programme on the young mothers’ parenting self-efficacy and behaviour, and the growth and development of children under five. A detailed discussion will be presented in this section.

### 4.1. Effect of Parenting Peer Education Programme on the Young Mothers’ Parenting Self-Efficacy

The health education given to young mothers was expected to improve the growth and development of their babies because it can increase the mothers’ parenting self-efficacy, which was an important variable in adaptation to motherhood and maternal roles. The concept of Mutual Recognition Arrangement (MRA) nursing is one of the efforts which are made to improve mothers’ parenting self-efficacy (PSE) [50].

This study showed that the young mothers with the PPE programme reported significantly better PSE than the young mothers without the PPE programme (*p*-value = 0.000 and η^2^
*p* = 0.308). The finding of this study was in line with the findings of previous studies. Bloomfield and Kendall [51] reported that parents showed significant changes in both parenting stress and parenting self-efficacy after participating in the parenting programme [51]. At the beginning of the programme, many parents reported that they had a very high level of parenting stress, and this level normally remained until three months after attending the parenting programme. The parenting programme positively affected parental efficacy in increasing the children’s behaviours [27]. The parenting programme given to parents created a strong parental self-efficacy. Effective mindset and positive self-efficacy in parents will foster intrinsic interest and involvement in helping find ways of parenting their children [27].

Bandura [52] stated that self-efficacy is not a personality trait, but it is a component of beliefs which can be learned and produces a positive effect on the quality of parenting. It was found that peer education can significantly increase mothers’ self-efficacy in breastfeeding [53]. Peer education can be used as a solution to improve the mothers’ self-efficacy and continuity of exclusive breastfeeding [53]. This study also supported a previous study conducted in the University Hospital of Zurich in which the intervention programme is needed to improve the parenting self-efficacy during the pandemic period [54].

On the other hand, the finding of this study was contradictive to the finding of a previous study by Kartini et al. [55]. The previous study was conducted at a community health centre in Temanggung, Central Java, Indonesia and reported that there was no significant difference in the self-efficacy score between the control group and intervention group after the intervention of health education was given [55]. The different findings might be caused by some factors such as parenting methods, respondents’ basic characteristics, and educator qualifications.

### 4.2. Effect of Parenting Peer Education Programme on Young Mothers’ Parenting Behaviour

Parents’ knowledge and understanding of parenting behaviour are very influential on the children’s growth and development. Parents need to be given skills in educating children in the family and knowledge of caring for and guiding children so their children can become quality human resources in the future.

The PPE programme is one of the programmes that can be used to improve young mothers’ quality as parents in their families. The parenting peer education programme includes inculcating child-friendly parental attitudes or behaviours such as education-friendly, nutritional-friendly, care-friendly, and protection-friendly behaviours that are needed by their children and influence the children’s development phases in a structured and orderly manner. Therefore, intervention is needed to give the young mothers an understanding and facilitate adjustment in carrying out their role as parents [56].

This study showed that the young mothers who accessed the PPE programme reported significantly better parenting behaviour than the young mothers without the PPE programme (*p*-value = 0.015 and η^2^
*p* = 0.256). The young mothers who received the PPE programme had a higher parenting behaviour score compared to young mothers who did not receive the PPE programme. Parenting education is very much needed for parents because with parenting education parents can change their inappropriate parenting behaviour to be more appropriate. The parenting education is important not only for parents, but also for young people who are not yet parents, because the latter can learn it faster [57].

Conditions where teenagers are to be young mothers affect the parenting behaviour, which has an impact on the children’s growth and development. Young mothers usually have less knowledge about the importance of parenting for their children, and if this condition is coupled with low socio-economic conditions and low environmental support, parenting tends to be neglected. With these conditions, it is important to give attention to the efforts or support for young mothers in parenting their children. Support for young mothers through parenting peer education is an important thing that affects the children’s growth and development for the achievement of the welfare of the next generation in society [58].

### 4.3. Effect of Parenting Peer Education Programme on Children under Five’s Growth

Growth has an impact on physical aspects to achieve optimal growth depending on biological potential. The level of achievement of a child’s biological potential is the result of the interaction of various interrelated factors, namely genetic, environmental, parental, socio-cultural, nutrition, children and early disease detection factors. Therefore, it needs to be supported by many factors to ensure the optimal growth of children.

This study showed that the young mothers with the PPE programme reported significantly better growth of children under five than the young mothers without the PPE programme (*p*-value = 0.000 and η^2^
*p* = 0.307). Providing parenting peer education to parents can improve abilities and have a positive impact on parents to maximize their children’s growth. The provision of health education is an effective strategy to improve one’s health behaviour in terms of early detection of children’s growth. For this reason, good parental knowledge can be one of the supporting factors in supporting children’s growth [59]. Some previous studies reported the same finding with this study. A previous study in East Lombok, Indonesia showed that integrated nutrition rehabilitation interventions had benefits in increasing dietary diversity (in children <24 months) and weight-for-age children in post-disaster conditions [60]. Furthermore, a previous study by Yunarsih and Rahmawati [35] reported that there was an effect of the stimulation given to young mothers on the under fives’ growth [35]. Imdad et al. [61] also reported that the intervention of education for mothers regarding complementary foods for breast milk and complementary foods significantly increased the body weight and linear growth of children aged 6–24 months. The education about complementary foods for breast milk caused the children to gain weight and height faster in the intervention group than in the control group.

On the other hand, some previous studies reported different results. A previous study conducted in rural areas of Uganda [62] reported that there was no difference in children’s growth between the parenting intervention group and the non-intervention group [62]. Furthermore, a study by Muhoozi et al. [63] reported that the intervention of education on nutrition, hygiene, and stimulation for mothers did not correlate with the children’s growth [63]. Moreover, a previous study by Atukunda et al. [64] reported that there was no significant difference in the growth of children aged 60–72 months, which was shown by the no significant difference in anthropometric z-scores in the control group and intervention group [64]. It also showed no significant differences in body composition (fat mass, fat-free mass, and total body water) in children aged 60–72 months. In conclusion, there was no significant correlation between interventions and anthropometric z-scores, growth rate, height, and body composition of children.

### 4.4. The effect of Parenting Peer Education Programmes on Children under Five’s Development

The results of this study showed that, compared with the control group, the young mothers in the intervention group reported significantly better children under five’s cognitive development (*p* = 0.001, η^2^
*p* = 0.521) and children under five’s language development (*p* = 0.032, η^2^
*p* = 0.081). Meanwhile, the PPE programme resulted in a statistically significant difference in the under-fives’ motoric development between the two groups, but the effect size was very small (*p* = 0.046, η^2^
*p* = 0.008). A previous study in Sugihwaras Village, Prambon District, Nganjuk Regency, also reported that there was an effect of stimulation on mothers who married at a young age on the development of toddlers [35]. Research in Uganda reported that educational interventions on nutrition, hygiene and stimulation given to mothers resulted in significant improvements in children’s development, including cognitive, language and motoric aspects. The intervention group produced higher child development scores than the control group [35].

According to Kumar and Huang [65], it is important to provide integrated care to young mothers that correlates with health and care needs to optimize child development [65]. Singla et al. [62] also reported that there was a higher improvement in the cognitive and language scores of children in the parenting intervention group than children in the control group [62]. Furthermore, Khofiyah [66] reported that the development of stimulation education of mothers affected the children’s cognitive development score, but did not affect the children’s language and motoric development scores statistically [66]. Luoto et al. [67] stated that the parenting intervention was significantly related to children’s cognitive and motoric development but not to children’s language development [67]. The children living in the intervention village had higher cognitive and motoric scores compared to children living in the control village. However, the intervention did not affect the children’s language scores. Different timings of the intervention, the characteristics of the study area, the research method used, the number of samples analyses, and the model and method of intervention carried out might cause the difference in results between this study and the other studies.

The PPE programme can provide knowledge about the prenatal adaptation of the role of parents in children’s development, pregnancy, knowledge, and skills in parenting behaviour as well as additional coping skills in overcoming the psychological challenges of children’s development. In addition, the PPE programme can exchange the young mothers’ experiences on how their children grow and develop, and the mothers will also receive psychological support. This will lead to some changes in young mothers, including: (1) cognitive abilities such as thinking and mentality; (2) emotional strategies in perspective taking, empathy, ability to be sensitive and responsive to children’s cues, independence, and focus on the needs of children’s growth and development; and (3) situational factors regarding mental health, psychological preparation, parents’ roles, and social supports. Some of these factors can increase attitudes, subjective norms, belief in abilities, and self-efficacy towards parenting to reduce stunting.

The PPE can be used as a reference for the formulation of educational policies related to young mothers’ parenting self-efficacy and behaviour, and the growth and development of children under five, as well as the implementation of the PPE programme providing knowledge about PPE programmes to improve young mothers’ parenting self-efficacy and behaviour, and growth and development of children under five. In general, the PPE programme could be conducted elsewhere and could give similar findings to this study if the control variables (characteristics of young mothers, characteristics of children under five, and characteristics of peer educators) and the independent variable (the intervention model) are the same as those used by this study. The limitations in this study are the relatively small number of samples, the implementation of the intervention during the COVID-19 pandemic and the length of the intervention, which was just 3 months. Therefore, in future studies, we recommend using a larger sample, implementing strict interventions and extending the intervention time to obtain more accurate growth and development outcomes for children under five.

## 5. Conclusions

The results of this study showed that PPE significantly increased parenting self-efficacy, parenting behaviour in young mothers and children’s growth and development (cognitive, language and motor aspects). The results of this study and data from the literature review converge to maintain the importance of PPE programmes on PSE, parenting behaviour and children’s growth and development. In the future, the results of this research can be used as a reference for formulating educational policies for young mothers regarding parenting self-efficacy, parenting behaviour, and toddler growth and development, as well as the implementation of PPE programmes for young mothers who live around public health centres in Indonesia. This study was limited by the small sample size and the implementation of the research, which was carried out during the COVID-19 pandemic. Therefore, future research can use a larger sample to obtain more accurate results. In addition, increasing the duration of the intervention could be considered so that the impact on children’s growth and development can be measured more accurately.

## Figures and Tables

**Figure 1 children-10-00338-f001:**
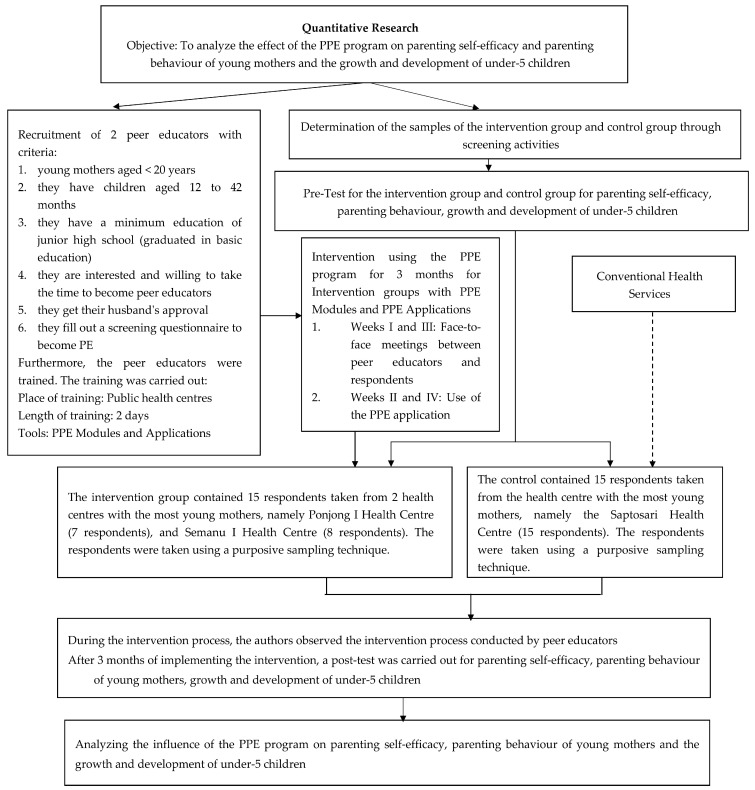
Detailed steps in this quantitative study.

**Table 1 children-10-00338-t001:** Summary of previous studies on parenting programmes for young mothers.

Parenting Programmes	Respondents	Effect of the Parenting Programmes on	References
Home visit by an eligible person on parenting interventions	Young mothers aged 12–18 years from 24 weeks pregnant to 3 years after giving birth	Young mothers’ parenting skill	[34]
Stimulation	Young mothers who have children under 5	Growth and development of children under 5	[35]
Mindful with Your Baby (MwyB)	Young mothers who have children aged 1–18 months	Increasing the efficacy of parents in parenting	[36]
Parenting Peer Education	Young mothers who have children under 5	Young mothers’ parenting self-efficacy Young mothers’ parenting behaviour Growth and development of children under 5	This study

**Table 2 children-10-00338-t002:** Baseline Equivalence between intervention group and control group (*n* = 30).

Variables	Intervention Group Mean ± SD	Control Group Mean ± SD	*p*-Value
Young mothers’ parenting self-efficacy score	79.4 ± 6.62	82.5 ± 7.03	0.8902
Young mothers’ parenting behaviour score	51.8 ± 3.95	50.27 ± 4.82	0.9533
Children under 5 ’s growth score	−0.43 ± 1.38	−0.81 ± 1.32	0.2208
Children under 5’s development score			
Cognitive	95.2 ± 11.74	90.67 ± 11.32	0.1454
Language	88.33 ± 10.09	82.73 ± 9.13	0.0611
Motoric	95.27 ± 9.69	97 ± 11.17	0.6733

Source: Baseline Data.

**Table 3 children-10-00338-t003:** The results of the data normality test.

Variables	Group								Difference *p*-Value
	Intervention				Control				
	Pre-Test	*p*-Value	Post-Test	*p*-Value	Pre-Test	*p*-Value	Post-Test	*p*-Value	
Young mothers’ parenting self-efficacy	15	0.973	15	0.899	15	0.884	15	0.974	0.958
Young mothers’ parenting behaviour	15	0.964	15	0.974	15	0.956	15	0.922	0.985
Children under 5’s growth	15	0.925	15	0.936	15	0.933	15	0.873	0.919
Children under 5’s development									
Cognitive	15	0.934	15	0.988	15	0.971	15	0.984	0.901
Language	15	0.942	15	0.791	15	0.985	15	0.966	0.949
Motoric	15	0.898	15	0.974	15	0.882	15	0.938	0.966

**Table 4 children-10-00338-t004:** The results of the homogeneity test.

Variables	Intervention Mean [SD]	Control Mean [SD]	*p*-Value
Young mothers’ parenting self-efficacy	79.4 [6.62]	82.5 [7.04]	0.919
Young mothers’ parenting behaviour	51.8 [3.95]	50.3 [4.82]	0.464
Children under 5’s growth	−0.4 [1.38]	−0.8 [1.32]	0.925
Children under 5’s development			
Cognitive	95.2 [11.74]	90.7 [11.32]	0.513
Language	88.3 [10.09]	82.7 [9.13]	0.716
Motoric	95.3 [9.69]	97.0 [11.17]	0.819

**Table 5 children-10-00338-t005:** The results of the ANCOVA with the pre-test scores as covariates.

Variables	F	*p*-Value	η^2^ *p*
Young mothers’ parenting self-efficacy	20.809	0.000	0.308 (large effect)
Young mothers’ parenting behaviour	5.953	0.015	0.256 (large effect)
Children under 5’s growth	103.797	0.000	0.307 (large effect)
Children under 5’s development			
Cognitive	14.012	0.001	0.521 (large effect)
Language	2.802	0.032	0.081 (medium effect)
Motoric	12.281	0.046	0.008 (very small effect)

Note: ANCOVA with post-test scores as dependent variables, intervention or control group as an independent variable, pre-test scores as covariates.

## Data Availability

Not applicable.

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
