# Peer review of "The Effect of Parenting Peer Education Interventions for Young Mothers on the Growth and Development of Children under Five"

_children, 2023, doi:10.3390/children10020338_

Round 1
Reviewer 1 Report
1. In the section of the introduction, I suggest the author should provide some subtitle to describe the main arguments and focused on the theoretical foundations and importance of the parenting peer education about this concern.
2. In the section of the method, I suggest the author should provide more explanations about the reasons why to employ the quasi-experimental research and how to use this method briefly.
3. In the section of the measurement, I suggest the author could provide some literature review based on these scales in the section of the introduction.
4. In the section of the results, the author provided well and robust statistical results to support the research hypotheses.
5. In the section of the conclusion, I suggest the author should provide some information about the theoretical reflections and practical suggestion based on the results and deleted the statistical results.
Author Response
|
NO |
Reviewer’s comments
|
Authors’ responses |
|
1 |
In the section of the introduction, I suggest the author should provide some subtitle to describe the main arguments and focused on the theoretical foundations and importance of the parenting peer education about this concern. |
Thank you. We have added many sentences about those. Please check Lines 52-66 Lines 119-142
|
|
2 |
In the section of the method, I suggest the author should provide more explanations about the reasons why to employ the quasi-experimental research and how to use this method briefly. |
Thank you. We have added information about those.Please check Lines 202-213 |
|
3 |
In the section of the measurement, I suggest the author could provide some literature review based on these scales in the section of the introduction. |
Thank you. We have added information about those. Please check Lines 146-152 |
|
4 |
In the section of the results, the author provided well and robust statistical results to support the research hypotheses. |
Thank you |
|
5 |
In the section of the conclusion, I suggest the author should provide some information about the theoretical reflections and practical suggestion based on the results and deleted the statistical results. |
Thank you. We have revised the conclusions. Please check Lines 587-599 |
Reviewer 2 Report
The theme of article is attractive, interesting and contain novelty. It would be great if following improvements can be considered before a final publication.
Firstly as a smaller issue, the title appears a bit long. It is just too many words in the current form for an article, and loses interest. Some possibilities could be simply being more direct and specific in whom the sample subject group is from the beginning and not trying to make it sound so universal. Sometimes when you put too many specific concepts in the title it excludes readers who do not know they may find the piece interesting as it looks too specialised and presumes the knowledge of words they never have yet seemed defined before.
Secondly, the abstract includes data and abbreviations for concepts that should actually only be spelled out in the body of the full article. Again, this is a trend in academia where there is a lack of proper editing and peer review, and needs to be fixed for publication for the reputation of the journal.
Thirdly, researchers should obviously not assume their location is known or is not relevant as context is influential, or let it ever appear they think their country is the whole world in an international journal where one hopes not to offend international readers through any kind of cultural imperialism, likely unintended of course but that is part of the issue. Readers should not have to guess, but should be told of the context and given a brief statement on its relevance.
Fourth, for similar reasons around internationalisation, the literature review should be citing a few more recent international studies to show the international nature of the publication and thinking of the authors a little more, with respect to readership diversity. On this aspect I recommend consulting the following articles:
Ansell, N. [2016], Children, Youth and Development, London, Taylor e Francis.
Elliott, S., Powell, R., e Brenton, J., Being a good mom: Low-income, Black single mothers negotiate intensive mothering, in «Journal of Family Issues», 2015, n. 36, pp. 351–370.
Fram, M. S. [2003], Managing to parent: Social support, social capital, and parenting practices among welfare-participating mothers with young children, Washington, Institute for Research on Poverty.
Golombok, S. [2016], Modern Families. Parents and Children in New Family Forms, Cambridge, Cambridge University Press.
Hays, S. [1996]. The Cultural Contradictions of Motherhood, New Haven, Yale University Press.
Ito, Y., e Izumi-Taylor, S., A comparative study of fathers’ thoughts about fatherhood in the USA and Japan, in «Early Child Development and Care», 2013, n. 183, pp. 1689–1704.
Jackson, D., e Mannix, J. [2004], Giving voice to the burden of blame: A feminist study of mothers’ experiences of mother blaming, in «International journal of nursing practice», 4, 2004, n. 10, pp. 150–158.
Jacquet, N., Van Haute, D., Schiettecat, T., e Roets, G., Stereotypes, conditions, and binaries: analysing processes of social disqualification towards children and parents living in precarity, in «British Journal of Social Work», 6, 2022, n. 52, pp. 3425–3442
Miller, W., e Maiter, S., Fatherhood and culture: Moving beyond stereotypical understandings, in «Journal of Ethnic & Cultural Diversity in Social Work», 3, 2008, n. 17, pp. 279-300.
Morgan, D. H. G. [1996], Family Connections: An Introduction to Family Studies, Cambridge, Polity.
Morgan, D. H. G. [2011], Rethinking family practices, Basingstoke, Palgrave Macmillan.
Fifth, it would be good in considering internationalisation to ensure a line about transferability issues in the limitations discussion at the end of the piece. In considering how to write this, one asks is it likely the findings sets hold true for people in Zimbabwe, New Zealand, Japan? Is this likely to occur say in Eastern countries but not other settings? Without too much detail, we need you to explain broadly how the study's context may or may not be similar to others in ways impacting transferability for some or all core findings as a general statement or two before the conclusion.
I think with these changes the piece would be a wonderful addition to the journal. These changes are very important but should be easily made.
Author Response
|
NO |
Reviewer’s Comments
|
Feed Back |
|
|
The theme of article is attractive, interesting and contain novelty. It would be great if following improvements can be considered before a final publication.
|
Thank you |
|
1 |
Firstly as a smaller issue, the title appears a bit long. It is just too many words in the current form for an article, and loses interest. Some possibilities could be simply being more direct and specific in whom the sample subject group is from the beginning and not trying to make it sound so universal. Sometimes when you put too many specific concepts in the title it excludes readers who do not know they may find the piece interesting as it looks too specialised and presumes the knowledge of words they never have yet seemed defined before.
|
Thank you. We have revised the title.Please check Lines 2-3 |
|
2 |
Secondly, the abstract includes data and abbreviations for concepts that should actually only be spelled out in the body of the full article. Again, this is a trend in academia where there is a lack of proper editing and peer review, and needs to be fixed for publication for the reputation of the journal.
|
Thank you. We have revised the abstract.Please check Lines 21-25 |
|
3 |
Thirdly, researchers should obviously not assume their location is known or is not relevant as context is influential, or let it ever appear they think their country is the whole world in an international journal where one hopes not to offend international readers through any kind of cultural imperialism, likely unintended of course but that is part of the issue. Readers should not have to guess, but should be told of the context and given a brief statement on its relevance. |
Thank you. We have revised the sentences. Please check Lines 52-66. |
|
4 |
Fourth, for similar reasons around internationalisation, the literature review should be citing a few more recent international studies to show the international nature of the publication and thinking of the authors a little more, with respect to readership diversity. On this aspect I recommend consulting the following articles: Ansell, N. [2016], Children, Youth and Development, London, Taylor e Francis. Elliott, S., Powell, R., e Brenton, J., Being a good mom: Low-income, Black single mothers negotiate intensive mothering, in «Journal of Family Issues», 2015, n. 36, pp. 351–370. Fram, M. S. [2003], Managing to parent: Social support, social capital, and parenting practices among welfare-participating mothers with young children, Washington, Institute for Research on Poverty. Golombok, S. [2016], Modern Families. Parents and Children in New Family Forms, Cambridge, Cambridge University Press. Hays, S. [1996]. The Cultural Contradictions of Motherhood, New Haven, Yale University Press. Ito, Y., e Izumi-Taylor, S., A comparative study of fathers’ thoughts about fatherhood in the USA and Japan, in «Early Child Development and Care», 2013, n. 183, pp. 1689–1704. Jackson, D., e Mannix, J. [2004], Giving voice to the burden of blame: A feminist study of mothers’ experiences of mother blaming, in «International journal of nursing practice», 4, 2004, n. 10, pp. 150–158. Jacquet, N., Van Haute, D., Schiettecat, T., e Roets, G., Stereotypes, conditions, and binaries: analysing processes of social disqualification towards children and parents living in precarity, in «British Journal of Social Work», 6, 2022, n. 52, pp. 3425–3442 Miller, W., e Maiter, S., Fatherhood and culture: Moving beyond stereotypical understandings, in «Journal of Ethnic & Cultural Diversity in Social Work», 3, 2008, n. 17, pp. 279-300. Morgan, D. H. G. [1996], Family Connections: An Introduction to Family Studies, Cambridge, Polity. Morgan, D. H. G. [2011], Rethinking family practices, Basingstoke, Palgrave Macmillan.
|
Thank you. We have added information in the body text with citing some articles that you recommend. Please check Lines 86-96Lines 462-467 |
|
5 |
Fifth, it would be good in considering internationalisation to ensure a line about transferability issues in the limitations discussion at the end of the piece. In considering how to write this, one asks is it likely the findings sets hold true for people in Zimbabwe, New Zealand, Japan? Is this likely to occur say in Eastern countries but not other settings? Without too much detail, we need you to explain broadly how the study's context may or may not be similar to others in ways impacting transferability for some or all core findings as a general statement or two before the conclusion.
|
Thank you. We have added statements.Please checkLines 562-565Lines 582-585 |
|
|
I think with these changes the piece would be a wonderful addition to the journal. These changes are very important but should be easily made.
|
Thank you |